# Evaluation of the Impact of Two Thiadiazole Derivatives on the Dissolution Behavior of Mild Steel in Acidic Environments

**DOI:** 10.3390/molecules28093872

**Published:** 2023-05-04

**Authors:** Ibrahim O. Althobaiti, Salah Eid, Karam S. El-Nasser, Nady Hashem, Eid Eissa Salama

**Affiliations:** 1Department of Chemistry, College of Science and Arts, Jouf University, Qurayyat 75911, Saudi Arabia; salaheed@ju.edu.sa (S.E.); karamsaif@ju.edu.sa (K.S.E.-N.); nhm00@fayoum.edu.eg (N.H.); 2Chemistry Department, Faculty of Science, Benha University, Benha 13518, Egypt; 3Department of Chemistry, Faculty of Science, Al-Azhar University, Assiut 71524, Egypt; 4Chemistry Department, Faculty of Science, Fayoum University, Fayoum 63514, Egypt; 5Chemistry Department, Faculty of Science, Suez Canal University, Ismailia 41522, Egypt

**Keywords:** corrosion inhibition, mild steel, thiadiazole, DFT

## Abstract

In light of the variety of industrial uses and economic relevance of mild steel, corrosion resistance is a serious topic. Utilization of inhibitors serves as one of the most essential methods for corrosion control. Two thiadiazole compounds, namely, 2-amino-5-(4-bromobenzyl)-1,3,4-thiadiazole (**a1**) and 2-amino-5-(3-nitrophenyl)-1,3,4-thiadiazole (**a2**), were synthesized. The structure of the prepared compounds was verified by Fourier transform infrared spectroscopy (FTIR) and proton and carbon-13 nuclear magnetic resonance spectroscopy (^1^H NMR and ^13^C NMR). In a 0.50 M H_2_SO_4_ solution, the effectiveness of two synthetic thiadiazole derivatives as mild steel corrosion inhibitors were investigated. In this evaluation, various electrochemical methodologies have been utilized, such as potentiodynamic polarization, open circuit potential (OCP), and electrochemical impedance spectroscopy (EIS). The results confirm the efficiency of the inhibition increases by raising concentrations of **a1** and **a2**. The inhibitory behavior was explained by the notion that the adsorption of thiadiazole molecules, **a1** and **a2,** on the surface of mild steel causes a blockage of charge and mass transfer, protecting the mild steel from offensive ions. Furthermore, the synthesized molecules **a1** and **a2** were analyzed using density functional theory (DFT).

## 1. Introduction

Mild steel (M-steel) is an economical and frequently utilized construction material in a broad range of industries due to its cost-effectiveness, low carbon content, and superior mechanical performance. Mild steel is susceptible to severe corrosion in acidic media [1,2,3,4,5,6,7,8]. Acidic solutions, particularly H_2_SO_4_ acid solutions, are commonly utilized in both chemical and industrial processes [9,10]. Inhibitors are among the most commonly utilized practical strategies for maintaining metals in corrosive environments [11,12,13]. Several organic molecule inhibitors possessing heteroatoms such as nitrogen, sulfur, oxygen, or, various links such as amine, alcoholic, and carbonyl groups work through adsorption on the surface of metal [14,15]. Corrosion cannot harm the metal because the inhibitor coating adsorbs the entire corroded metal surface. The surface charge and origin of metallic materials, the chemical composition of corrosion inhibitors, and the type of aggressive solution all have an impact on the corrosion process [16]. 

When organic inhibitors contain specific halogen ions, their effectiveness can be increased. Furthermore, it is known that in acidic conditions, halogen ions can partially prevent corrosion. The order of corrosion inhibition effectiveness from most effective to least effective is iodide, bromide, and chloride ions. Fluoride does not possess any features that function in corrosion inhibition. The metal adsorbs halogen ions, whose charge move the surface in a negative direction, causing synergism of halogen ions which increases adsorption to the cationic organic inhibitor. It takes a lot of effort, creativity, and laboratory analysis/synthesis to identify potential chemicals that can be employed as corrosion inhibitors [17,18,19].

Many researchers have utilized numerous organic heterocyclic compounds as anti-corrosion agents. In particular, heterocyclic molecules with sulfur, oxygen, and nitrogen atoms, for example, thiazole, oxazole, and azole derivatives, have shown significant beneficial efficiency against metallic corrosion in various conditions [20]. 

In comparison to compounds containing only nitrogen or sulfur atoms, heterocycle-based inhibitors, which contain both elements in their structure, provide exceptional corrosion-preventive effects. In this respect, a number of nitrogen- and sulfur-containing azole compounds, such as thiazole and thiadiazole derivatives, have been shown to be effective inhibitors of corrosion on several metallic materials in many different corrosive conditions. According to the bulk of the literature studying corrosion, thiadiazole-based compounds are more important than thiazole-based compounds. This trend in attention is based on the idea that adding more heteroatoms (nitrogen atoms) to such heterocyclic compounds can increase their adsorption to the metal surface and hence increase the efficiency of their inhibition [21,22,23].

Several applications of 1,3,4-thiadiazoles in industries such as pharmaceuticals and agrochemicals have been thoroughly investigated. They act as inhibitor enhancers and decrease metal degradation created by their surroundings. Several 1,3,4-thiadiazole derivatives have been shown to be efficient anti-corrosion agents in various conditions. This is likely due to the higher orientation of heteroatoms with conjugated multiple bonds, which facilitates the adsorption of such compounds onto the metal surface; after that, a protective layer is formed that isolates the substrate from components of the solution [24,25].

It became clear that 1,3,4-thiadiazole derivatives’ ability to prevent metal dissolution is affected by the shape and size of the attached substituents as well as their chemical characteristics [26,27].

Our research aims to prepare and examine the inhibitory effect of two thiadiazole compounds, namely, 2-amino-5-(4-bromobenzyl)-1,3,4-thiadiazole (**a1**) and 2-amino-5-(3-nitrophenyl)-1,3,4-thiadiazole (**a2**), on the corrosion behavior of mild steel in H_2_SO_4_ medium utilizing potentiodynamic polarization, open circuit potential, and impedance spectroscopy methods. In addition, DFT was used to examine the two inhibitors, **a1** and **a2**, in order to better comprehend the mechanism of corrosion inhibition.

## 2. Results and Discussion

### 2.1. Synthesis

4-Bromophenylacetic acid or 3-nitrobenzoic acid were reacted with thiosemicarbazide in the presence of phosphorus oxychloride. The product was basified with potassium hydroxide to give 2-amino-5-(4-bromobenzyl)-1,3,4-thiadiazole (**1**) as white crystals in a 65% yield or 2-amino-5-(3-nitrophenyl)-1,3,4-thiadiazole (**2**) as yellow crystals in a 60% yield (*c.f* Table 1). 

The structure of 2-amino-5-(4-bromobenzyl)-1,3,4-thiadiazole (**1**) was confirmed on the basis of its spectral data through mass spectrometry which showed well defined fragments (m/e, % of relative abundance): 271.9 ([M + 2]^⨥^, 98), 269 (M^⨥^, 100) indicating the molecular formula C_9_H_8_BrN_3_S (Figure 1a). The ^1^H NMR spectrum (DMSO-*d*_6_) of the ABT compound exhibited all the expected peaks corresponding to the proposed structure. The primary amine –NH_2_ signal appeared at 7.038 ppm. The two protons of CH_2_ gave a signal in 4.107 ppm. The two symmetrical protons in CH at positions 2 and 6 in the phenyl ring gave rise to doublet of doublets (dd) in 7.225 ppm. The other two symmetrical protons in CH at positions 3 and 5 in the phenyl ring gave rise to doublet of doublets (dd) in 7.508 ppm (Figure 1b). The ^13^C NMR spectrum (DMSO-*d*_6_) of compound **1** exclusively confirmed the proposed structure. Two aromatic carbons at positions 2 and 5 in the thiadiazole ring showed two signals at 169.0 ppm and 157.3 ppm, respectively. The carbon of the methylene group between the phenyl ring and the thiadiazole ring showed a signal at 35.2 ppm. Six carbons at positions 1, (2, 6), (3, 5), and 4 in the phenyl ring showed four signals at 137.9, 131.4, 132.0, and 120.5, ppm respectively (Figure 1c).

Spectral data confirmed the structure of 2-amino-5-(3-nitrophenyl)-1,3,4-thiadiazole (**2**), as the mass spectrum of **2** showed the following fragment (m/e, % of relative abundance): 223 (M^⨥^, 100), indicating the molecular formula C_8_H_6_N_4_O_2_S (Figure 2a). The ^1^H NMR spectrum (DMSO-*d_6_*) of compound **2** exclusively confirmed the proposed structure. The primary amine -NH_2_ signal appeared around 7.62 ppm. The proton in CH at position 2 in the phenyl ring gave rise to a singlet at 8.45 ppm. The proton in CH at position 4 in the phenyl ring gave rise to a doublet at 8.22 ppm. The proton in CH at position 5 in the phenyl ring gave rise to a multiplet in the range of 7.68–7.74 ppm. The proton in CH at position 6 in the phenyl ring gave rise to a doublet at 8.12 ppm (Figure 2b).

### 2.2. Open Circuit Potential Measurements

Figure 3a depicts the open circuit potentials (OCP) versus time curves for M-steel in 0.50 M H_2_SO_4_ in the absence and presence of different concentrations of the **a1** inhibitor. Based on a thorough examination of OCP vs. time curves, the presence of **a1** resulted in a positive change in the steady-state probability. The steady state was generally obtained within 10 min of electrode immersion in the experiment solutions. In addition, as the concentration of **a1** increased, the OCP became more positive. The open circuit potentials versus time for the M-steel corrosion both in the blank and inhibited solutions containing 0.002 M concentrations of **a1** and **a2** compounds in 0.50 M H_2_SO_4_ medium are depicted in Figure 3b. It was determined, based on Figure 3b, that the effect of **a1** is greater than that of **a2**. The Fe samples’ OCP versus time curves were almost straight, signifying the achievement of a steady-state potential [28,29,30].

### 2.3. Potentiodynamic Polarization

Potentiodynamic polarization curves for the M-steel corrosion both in the blank and inhibited solution containing 0.005 M concentrations of **a1** and **a2** compounds in 0.50 M H_2_SO_4_ medium are depicted in Figure 4.

The addition of the examined inhibitors, **a1** and **a2,** into the solution had little impact on the general form of the potentiodynamic curves, showing that **a1** and **a2** merely block the reaction sites (active sites) of the M-steel surface without variation in the cathodic and anodic reaction mechanisms.

Adsorption represents the first step toward corrosion prevention in neutral solutions. The damping effect is created by the inhibitor molecules adsorbing onto the active corrosion locations on the M-steel surface. The chemical composition of the inhibitor, as well as the type and charge of the degraded surface, influence the adsorption mechanism [31]. This is because the surface of the M-steel to be inhibited is usually oxide-free, enabling the inhibitor’s immediate proximity to delay the cathodic and/or anodic electrochemical processes. Although the adsorbed inhibitor does not completely cover the M-steel’s surface, it does occupy electrochemically active sites and reduce the intensity of either the cathodic or anodic reactions, or both. The rate of corroding will be reduced in relation to the extent to which the electrochemically active areas are covered by the adsorbing inhibitor. From Table 2, it was detected that the addition of **a1** to the sulfuric acid had a larger effect on the M-steel corrosion rate when compared to the **a2** inhibitor.

Figure 5 depicts the effect of increasing the concentration of **a1** on potentiodynamic polarization plots for M-steel corrosion in H_2_SO_4_. It was discovered that the cathodic (hydrogen evolution reaction) and anodic (metal dissolution) mechanisms were altered as a result of adding the inhibitor to the aggressive media [31]. Several electrochemical parameters were computed from the extrapolation of Tafel branches, including cathodic (*β*_c_) and anodic (*β*_a_) Tafel slopes, corrosion current density (*i*_cor_), and potential corrosion (*E*_cor_) and are presented in Table 3. The inhibitory performance (*IP*%) and surface coverage (*θ*) were calculated using the following equation. $IP\%=\left(1-\frac{i_{surf}}{i_{free}}\right)\times 100=\theta \times 100$, where, *i*_free_ and *i*_surf_ are the corrosion current density (*i*_corr_*)* without and with **a1** and **a2** compounds, respectively. Table 3 shows that the inhibitory capability rises as the concentration increases, reaching around 97.1% in the presence of 0.005 M of **a1**. Such behavior could be understood by an enhanced adsorption of **a1** molecules onto the M-steel/H_2_SO_4_ solution interface [32,33], which is encouraged by increasing the surface coverage (*cf*. Table 3) Furthermore, there is no significant change in the *E*_cor_ following the application of the investigated chemical. This indicates that **a1** compounds function as mixed form inhibitors [34,35].

Inhibition efficiency percentages of our constructed thiadiazole derivatives are also shown in Table 4, along with the percentage of inhibition efficiency for various organic compounds that have been chosen and used as effective corrosion inhibitors in various conditions. The PP measurements with low doses of thiadiazole derivatives were used to derive the inhibitory efficacy values listed in this table [36,37,38,39,40,41,42,43]. Our two newly created thiadiazole derivatives (**a1**) are more potent inhibitors than other chosen chemical derivatives, as shown in Table 4.

### 2.4. Adsorption Isotherm

Polarization measurements were utilized to evaluate surface coverage (*θ*) values for adsorption of various **a1** concentrations on the surface of M-steel. Many adsorption isotherms were graphically examined to identify the most appropriate adsorption isotherm for **a1** adsorption on the surface of M-steel [44], Figure 6. The Langmuir adsorption isotherm was identified to be the closest match, and it is derived from the subsequent equation. [44,45]: $C/\theta =\left(1/K\right)+C$, where *C* is the **a1** concentration and *K* is the equilibrium constant of adsorption. Figure 6 reveals the graphing of C/θ vs. C, which produces straight lines with almost unit slopes for **a1** with an intercept of 1/*K*. The relationship between the standard free energy of adsorption (ΔG°_ads_) and equilibrium constant of adsorption (K) is described by the next equation [29]: $K=\left(1/55.5\right)\exp\left(-\Delta G_{\mathrm{ads}}^{o}/\mathrm{TR}\right)$, where *T* refers to the absolute temperature, H_2_O molar concentration, and *R* refers to the gas constant. In 0.5 M H_2_SO_4_, the free energy of adsorption for adsorbed **a1** on the iron surface is 31 kJ·mol^−1^, while the equilibrium constant is 5000. The fact that ∆G^o^_ads_ is negative indicates that **a1** adsorbs spontaneously on the iron surface [45,46,47].

### 2.5. Electrochemical Impedance Study (EIS)

Figure 7a,b shows EIS plots for M-steel corrosion in both the blank and inhibited solutions containing 0.002 M quantities of **a1** and **a2** chemicals in 0.50 M H_2_SO_4_ medium. In this investigation, all of the processes implicated in the system’s electrical response were matched with an equivalent Randle CPE circuit model (see *inset* Figure 7a). Where the resistance of the solution (Rs) describes ohmic resistance, the charge transfer resistance (Rct) represents the inhibitor’s resistance to metal surface oxidation and is inversely proportional to the rate of corrosion. A constant phase element (CPE) replaces a pure double layer capacitor (C_dl_) to justify the semicircle form of the Nyquist plot [48]. The larger the diameter of the Nyquist plot, the larger the Rct value, and consequently, the better the inhibitory efficacy of a particular inhibitor.

According to Figure 7, the results of these experiments show that the film resistances for M-steel have maximum values in the presence of **a1.** The impact of rising **a1** concentration on EIS plots for M-steel corrosion in a 0.50 M H_2_SO_4_ environment is shown in Figure 8. In general, raising the inhibitor concentration causes the observed impedance values to rise, indicating that the electrode surface becomes more passive. This could be related to the strengthening of the inhibitive layer on the mild steel surface [49]. The Bode plot figures show, at the intermediate frequency, one maximum phases. According to ohmic law, the greater the resistance value (Rct), the smaller the electrical current (I) flow, and hence the fewer the number of electrons that are transmitted over the surface of M-steel. Consequently, the metal dissolving process (iron oxidation) appears to be impeded [50]. The fact that the Nyquist plots retained semicircle forms throughout the studies demonstrated that corrosion protection occurs via a charge transfer mechanism [51,52]. The results of the impedance measurements were in good agreement with those of the potentiodynamic polarization experiments.

### 2.6. Theoretical Studies

Quantum chemical calculation is a molecular structure-based approach to problem solving that involves the computation of molecular parameters that have a substantial correlation with predictable response functions such as inhibition effectiveness. The thiadiazole derivatives under study had their geometry optimized at the DFT level using the functional B3LYP and basis set 6–31 G (d,p) using BIOVA Materials Studio 7.0 (Accelrys Inc., San Diego, CA, USA). According to reports, HOMO orbitals provide electrons, whereas LUMO orbitals accept electrons [53,54]. Figure 9 and Figure 10 display the optimized molecular structures of the **a1** and **a2** compounds with the HOMO and LUMO electronic densities that were obtained from theoretical calculations. The HOMO orbitals surrounding the benzyl ring on the **a2** molecule, according to the results from Figure 9 and Figure 10. For **a1**, the HOMO orbitals are dispersed throughout the entire molecule. On the other hand, the LUMO orbitals are located on the thiadiazole ring group for **a2** and over the entire molecule for **a1**. This demonstrates the molecules’ electron-donating center. The results of DFT demonstrate that **a1** and **a2** have various electron donor and acceptor sites. This also explains the difference in inhibition effectiveness between **a1** and **a2**. The theoretical computations produced findings that are compatible with the practical results.

## 3. Experimental

### 3.1. Materials

The working electrode utilized in the present work is mild steel (M-steel) embedded in epoxy holders with an exposed area of 0.12 cm^2^. The chemicals used in our investigation include Thiosemicarbazide, 4-bromophenyl acetic acid, 3-nitro benzoic acid, phosphorus oxychloride, phosphorus oxychloride, potassium hydroxide, and ethanol. All chemical reagents were purchased from Sigma-Aldrich (St. Louis, MO, USA).

### 3.2. Equipment and Instrumentation 

All melting points are given uncorrected in °C. They were computed on a MEL-TEMP II melting point instrument. The ^1^H NMR spectra of the compounds produced in this study were measured using a Bruker (300 MHz) spectrometer, while their IR spectra in KBr were measured using a Pye Unicam SP 1200 spectrometer. 

### 3.3. Synthesis of 1,3,4-Thiadiazole Derivatives

#### 3.3.1. Synthesis of 2-Amino-5-(4-Bromobenzyl)-1,3,4-Thiadiazole 

Thiosemicarbazide (0.455 g, 1 mmole) and 4-bromophenyl acetic acid (1 mmole) were poured in phosphorus oxychloride (3 mL). The mixture mixed for 1 h and then it was cooled after which 10 mL of water was added. The mixture was heated for three hours, followed by filtration. Potassium hydroxide was added until the solution was basic. The precipitate was recrystallized from ethanol.

#### 3.3.2. Synthesis of 2-Amino-5-(3-Nitrophenyl)-1,3,4-Thiadiazole

Thiosemicarbazide (0.455 g, 1 mmole) and 3-nitro benzoic acid (1 mmole) were poured into phosphorus oxychloride (3 mL). The mixture was mixed for 1 h and cooled thereafter. Subsequently, 10 mL of water was added. The mixture was heated for three hours, followed by filtration. Potassium hydroxide was added until the solution was basic. The precipitate was recrystallized from ethanol.

### 3.4. Electrochemical Measurements

The electrochemical corrosion studies were carried out at room temperature using Versa STAT 4 and the Versa Studio Electrochemical software suite. The working, reference, and auxiliary electrodes in the three-electrode system in the jacketed glass cell were made of mild steel, calomel, and Pt, respectively. The working electrode was attached using epoxy resin, with 0.12 cm^2^ of its surface exposed to the tested hostile media. Before each run, the exposed region was polished to a mirror-like surface using grades up to 2500 grit. After 15 min of immersion in the test solution, the electrochemical tests were performed. At an open-circuit potential (OCP), potentiostatic circumstances were used for the EIS measurements, which covered the frequency range from 100 kHz to 0.1 Hz at an amplitude of 10 mV. The potentiodynamic polarization responses were investigated in a 0.5 M H_2_SO_4_ solution at 25 °C and a scan rate of 5 mV/s in without any corrosion inhibitors or in the presence of different quantities of corrosion inhibitors (0.0005, 0.001, 0.002, 0.003, 0.004, and 0.005 M). 

## 4. Conclusions

Two thiadiazole compounds, **a1** and **a2**, were synthesized and their chemical structures identified. Both **a1** and **a2** demonstrated the ability to create a protective coating on the M-steel surface against acidic media. 2-amino-5-(4-bromobenzyl)-1,3,4-thiadiazole (**a1**) demonstrates higher inhibition efficiency than 2-amino-5-(3-nitrophenyl)-1,3,4-thiadiazole for M-steel, and the inhibition efficiency increased as a function of inhibitor concentration. Surprisingly, a negative $\Delta G_{ads}^{0}$ implies that adsorption occurs spontaneously. The synthesized thiadiazole compounds adsorb on the surface of M-steel through a combination of chemical and physical processes that correlate to the Langmuir adsorption isotherm. The corrosion potential values did not change significantly, indicating that compounds **a1** and **a2** serve as mixed-type inhibitors. Theoretical studies based on DFT agree with our research observations demonstrating the efficacy of **a1** and **a2** as corrosion additives.

## Figures and Tables

**Figure 1 molecules-28-03872-f001:**
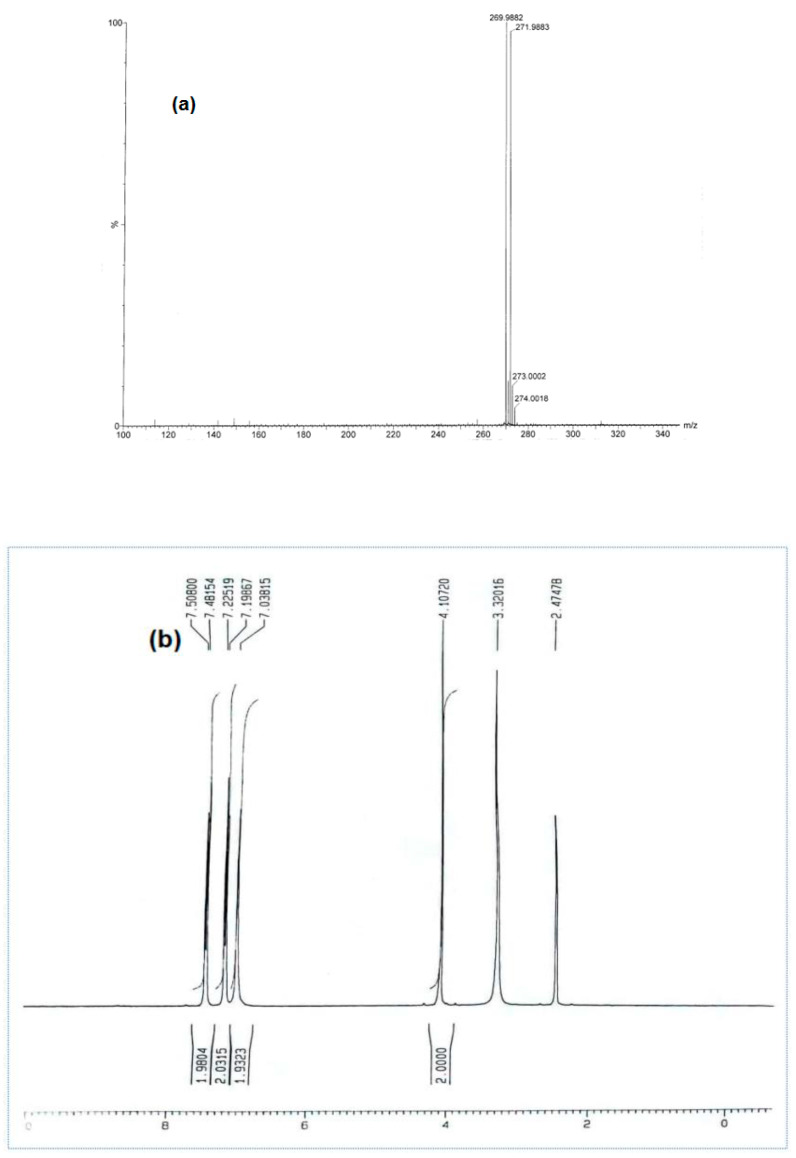
(**a**) Mass spectrometry, (**b**) ^1^H NMR spectrum, and (**c**) ^13^C NMR spectrum of 2-amino-5-(4-bromobenzyl)-1,3,4-thiadiazole.

**Figure 2 molecules-28-03872-f002:**
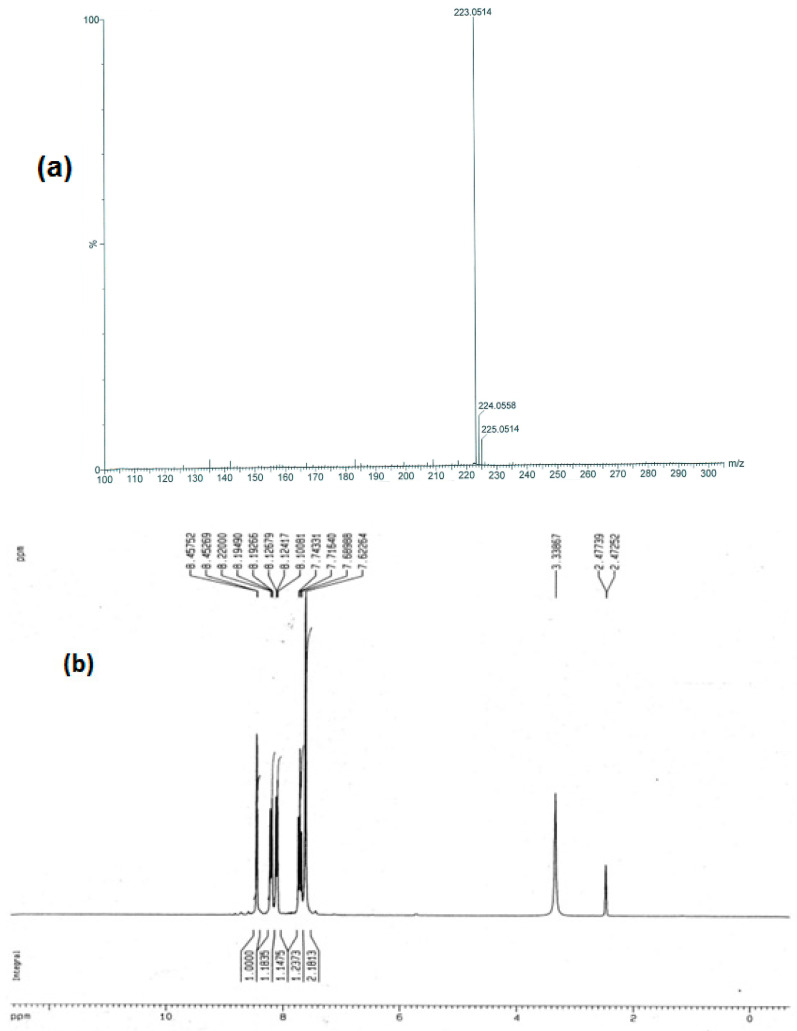
(**a**) Mass spectrometry and (**b**) 1HNMR spectrum of 2-amino-5-(3-nitrophenyl)-1,3,4-thiadiazole.

**Figure 3 molecules-28-03872-f003:**
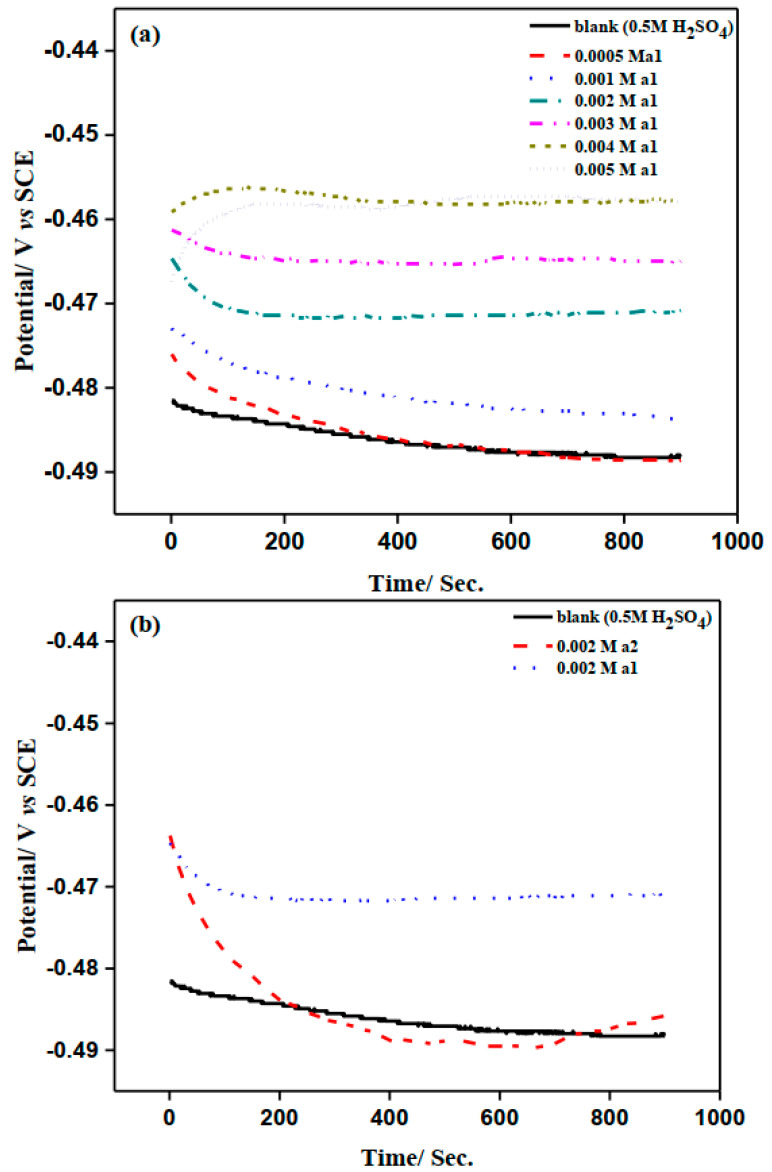
(**a**) OCP of mild steel electrode in the absence and presence of different concentration of 2-amino-5-(4-bromobenzyl)-1,3,4-thiadiazole. (**b**) OCP curves of mild steel in the absence and presence of 0.002 M of 2-amino-5-(4-bromobenzyl)-1,3,4-thiadiazole (**a1**) and 2-amino-5-(3-nitrophenyl)-1,3,4-thiadiazole (**a2**) in stagnant 0.5 M H_2_SO_4_ at 298 K.

**Figure 4 molecules-28-03872-f004:**
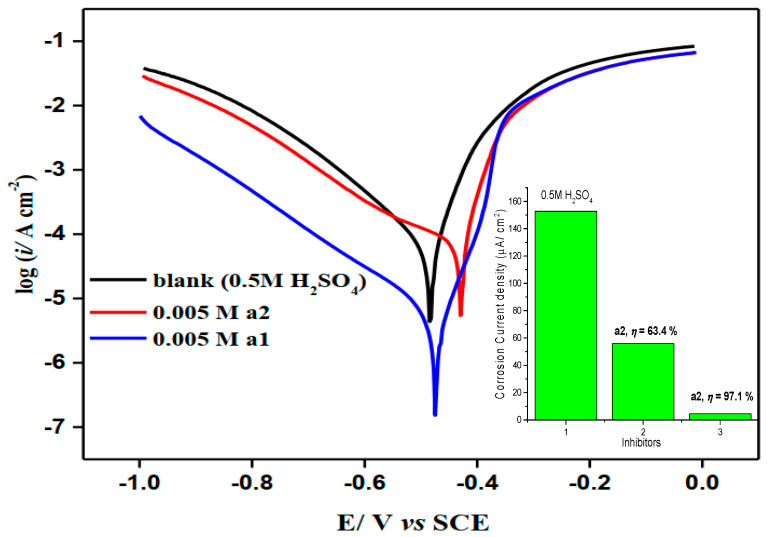
LogI–E curves of mild steel in the absence and presence of 0.002 M of 2-amino-5-(4-bromobenzyl)-1,3,4-thiadiazole and 2-amino-5-(3-nitrophenyl)-1,3,4-thiadiazole in stagnant H_2_SO_4_ (0.5 M) at 298 K.

**Figure 5 molecules-28-03872-f005:**
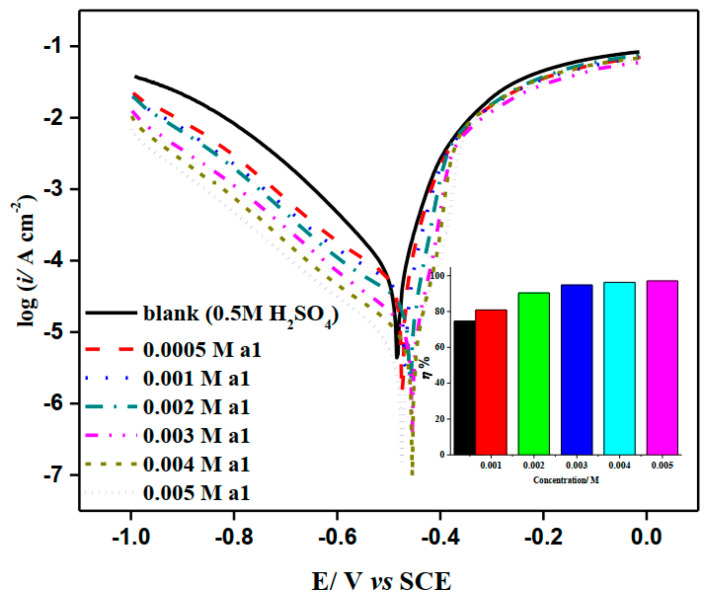
LogI–E curves of mild steel in the absence and presence of different concentrations of 2-amino-5-(4-bromobenzyl)-1,3,4-thiadiazole compound in stagnant 0.5 M H_2_SO_4_ at 298 K.

**Figure 6 molecules-28-03872-f006:**
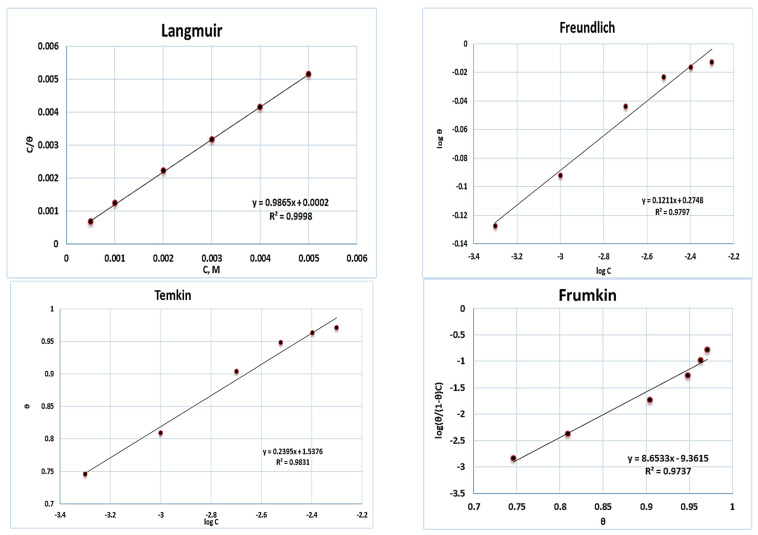
Adsorption isotherm plots for the adsorption of 2-amino-5-(4-bromobenzyl)-1,3,4-thiadiazole to mild steel in 0.5 M H_2_SO_4_.

**Figure 7 molecules-28-03872-f007:**
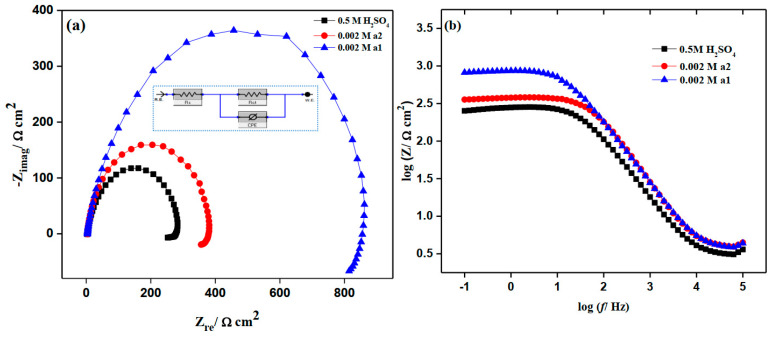
EIS curves of mild steel in the absence and presence of 0.002 M 2-amino-5-(4-bromobenzyl)-1,3,4-thiadiazole and 2-amino-5-(3-nitrophenyl)-1,3,4-thiadiazole in stagnant 0.5 M H_2_SO_4_ at 298 K. (**a**) Nyquist plots and (**b**) Bode plots.

**Figure 8 molecules-28-03872-f008:**
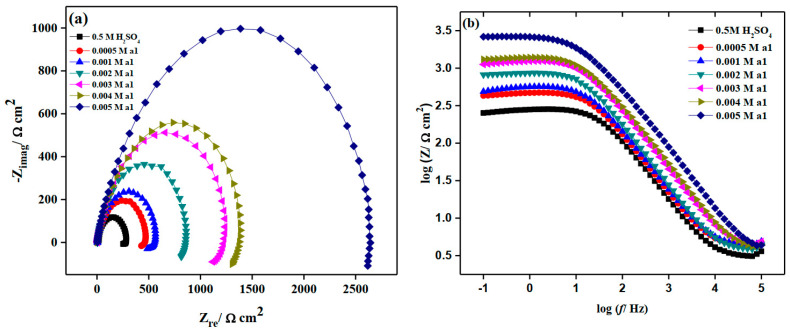
EIS curves of mild steel in the absence and presence of different concentrations of M 2-amino-5-(4-bromobenzyl)-1,3,4-thiadiazole in stagnant 0.5 M H_2_SO_4_ at 298 K. (**a**) Nyquist plots and (**b**) Bode plots.

**Figure 9 molecules-28-03872-f009:**
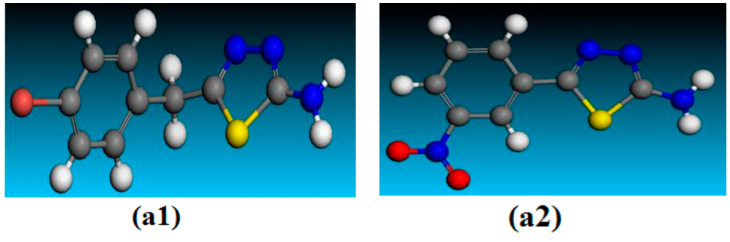
Optimized molecular structures of the studied thiadiazole derivatives **a1** and **a2** at DMol3/GGA.

**Figure 10 molecules-28-03872-f010:**
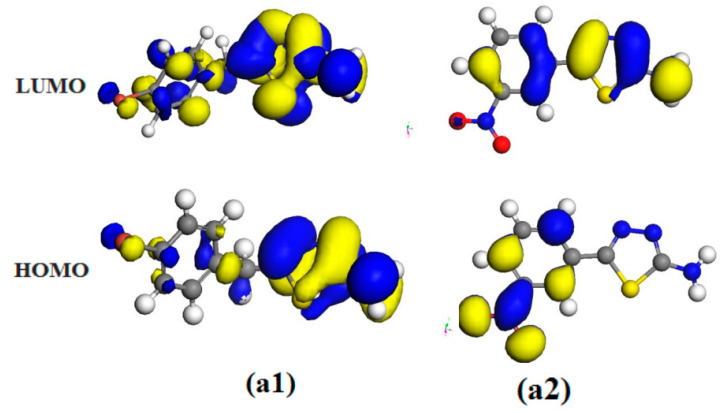
HOMO and LUMO for optimized structures of the studied thiadiazole derivatives **a1** and **a2** atDMol3/GGA.

**Table 1 molecules-28-03872-t001:** Molecular names, structures, and chemical formulae of the two thiadiazole derivatives under investigation as corrosion inhibitors.

Molecular Code	Molecular Structure	Molecular Name, Molecular Weight, Chemical Formula	Analysis
**a1**	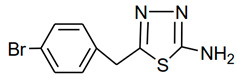	2-amino-5-(4- bromobenzyl)- 1,3,4-thiadiazole **Molecular weight:** 268.96 **Chemical Formula:** C_9_H_8_BrN_3_S	**White crystal, Yield 65%; mp: 200–202 °C; ^1^H NMR (300 MHz, DMSO-d6, **δ** ppm): 4.10720 (s, 2H,-CH_2_-), 7.03815 (s, 2H, -NH_2_), 7.22519 (d, 2H, J = 8.18 Hz), 7.50800 (d, 2H, *J* = 8.079 Hz); ^13^C NMR (DMSO-*d*_6_, **δ** ppm): (169.4, 157.3, 137.9, 132.0, 131.4, 120.5, 35.2); ESI-MS: 269.9 (100), 271.9 (98).**
**a2**	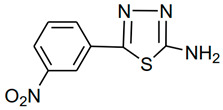	2-amino-5-(3- nitrophenyl)-1,3,4- thiadiazole **Molecular weight:** 222.02 **Chemical Formula:** C_8_H_6_N_4_O_2_S	Yellow crystal. Yield 60%, mp:236- 238 °C. ^1^H NMR (300 MHz, DMSO-*d*_6_, δ ppm): 7.62254 (s, 2H, -NH2), 7.71640–8.45752 (m, 4H, ArH); ESI-MS: 223 (100)

**Table 2 molecules-28-03872-t002:** Polarization parameters of the mild steel after electrode immersion in 0.5 M H_2_SO_4_ solutions in the absence and presence of 0.005M of **a1** and **a2**.

Inhibitor	Ecorr /mV	Icorr/ µA cm^−2^	ßa/ mV dec^−1^	ßc/ mV dec^−1^	θ	IP%
Free	−503.1	152.7	91.2	−174.4	-	-
0.005 M **a1**	−459.4	4.5	41.9	−169.7	0.971	97.1
0.005 M **a2**	−434.7	55.9	39.1	−201.6	0.634	63.4

**Table 3 molecules-28-03872-t003:** Polarization parameters of the mild steel after electrode immersion in a 0.5 M H_2_SO_4_ solution in the absence and presence of different concentrations of **a1**.

Inhibitor Conc/M	Ecorr /mV	Icorr / µA cm^−2^	ßa/ mV dec^−1^	ßc/ mV dec^−1^	θ	IP%
Free	−503.1	152.7	91.2	−174.4	-	-
0.0005	−487.7	38.8	52.1	−166.2	0.746	74.6
0.001	−478.5	29.1	44.7	−172.2	0.809	80.9
0.002	−454.6	14.7	29.3	−163.5	0.904	90.4
0.003	−434.7	8.0	23.5	−172.5	0.948	94.8
0.004	−443.3	5.6	25.4	−168.6	0.963	96.3
0.005	−459.4	4.5	41.9	−169.7	0.971	97.1

**Table 4 molecules-28-03872-t004:** Comparison of the inhibition effectiveness of various inhibitors in different media.

Inhibitor	Concentration	Inhibition Efficiency	References
5((6-Methyl-2-oxo-2*H*-chromen-4yl)thiomethyl)-2((*N*-(3-methylquinoxalin-2(1*H*)one)yl) methyl)-1,3,4-oxadiazole	50 ppm	84.85	[36]
1-(2-ethylamino-1,3,4-thiadiazol-5-yl)-3-phenyl-3-oxopropan (ETO)	500 ppm	98.4	[37]
*N*-cyanoacetohydrazide (CAH)	500 ppm	48.24	[38]
*N*-acryloylN0-cyanoacetohydrazide (ACAH)	500 ppm	91.1	[38]
poly(*N*-acryloyl-N0-cyanoacetohydrazide) (PACAH)	500 ppm	96.57	[38]
1-Amino-2-mercapto-5-(4-(pyrrol-1-yl)phenyl)-1,3,4-triazole (AMPPT)	500 ppm	91.4	[39]
*N*-trioctyl-*N*-methyl ammonium methylsulfate	100 ppm	92.0	[40]
*N*-tetradecyl-*N*-trimethyl ammonium methylsulfate	100 ppm	94.0	[40]
1-(4-sulfonic acid) butyl-3-ethyl imidazolium hydrogen sulfate	15 mM	88.0	[41]
1,10 (1,4 phenylenebis (methylene))bis (3 (carboxymethyl) 1*H* imidazole 3 ium) chloride com KI (1:4)	1 mM	88.5	[42]
1-butyl-1-methyl-pyrrolidinium Imidazolate	5 × 10^−3^ M	92.32	[43]
1-butyl-3-methyl-imidazolium Imidazolate	5 × 10^−3^ M	94.33	[43]
2-amino-5-(4-bromobenzyl)-1,3,4-thiadiazole (**a1**)	0.005 M	97.1	Our work
2-amino-5-(3-nitrophenyl)-1,3,4-thiadiazole (**a2**)	0.005 M	63.4	Our work

## Data Availability

Not applicable.

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
