# Peer review of "Evaluation of the Impact of Two Thiadiazole Derivatives on the Dissolution Behavior of Mild Steel in Acidic Environments"

_molecules, 2023, doi:10.3390/molecules28093872_

Round 1

Reviewer 1 Report

The paper titled “Evaluation impact of two thiadiazole derivatives on dissolution behavior of mild steel in acidic environments” is interesting and recommended for publications after the following corrections.

1.     Please check the grammatical error throughout the manuscript.

2.     Figure 4, corrosion efficiency of a1 is very large but a2 is very low lower than blank, Then no use of inhibitor why should you shown that result?

3.     Author not determined the weightloss method why? Better compare with Tafel, EIS etc

4.     Conclusion is not clear and must be reedited?

5.     The authors should give examples in:  Introduction of the formation of oxides by inhibition method using organic triazoles: cite these articles

K. Raviprabha, R.S. Bhat, Surf. Eng. Appl. Electrochem. 55 (6) (2019) 723–733.

K. Raviprabha, R.S. Bhat, J. Fail. Anal. Prev. 19 (2019) 1464–1474.

Overall good

Author Response

Reviewer #1

Dear reviewer,

Thank very much for your valuable and interesting comments which shall improve the manuscript.

The paper titled “Evaluation impact of two thiadiazole derivatives on dissolution behavior of mild steel in acidic environments” is interesting and recommended for publications after the following corrections.

  1. Please check the grammatical error throughout the manuscript.

Authors’ response: The English was polished carefully and the necessary revision and correction was made and highlighted in yellow.

  1. Figure 4, corrosion efficiency of a1 is very large but a2 is very low lower than blank, Then no use of inhibitor why should you shown that result?

Authors’ response: To investigate the effect of some functional groups on the inhibition efficiency of molecules.

  1. Author not determined the weightloss method why? Better compare with Tafel, EIS etc.

Authors’ response: In this research we use different electrochemical techniques to study the impact of two thiadiazole derivatives on dissolution behavior of mild steel in acidic environments.

All these techniques support that investigated inhibitors act as a good inhibitor for carbon steel corrosion in sulfuric acid.

  1. Conclusion is not clear and must be reedited?

Authors’ response: The conclusions were improved

  1. The authors should give examples in:  Introduction of the formation of oxides by inhibition method using organic triazoles: cite these articles
  2. Raviprabha, R.S. Bhat, Surf. Eng. Appl. Electrochem. 55 (6) (2019) 723–733.
  3. Raviprabha, R.S. Bhat, J. Fail. Anal. Prev. 19 (2019) 1464–1474

Authors’ response: the comment was addressed

Reviewer 2 Report

This manuscript must be largely corrected. I pointed out some small or severe problems in the manuscript, as follows.

The resolution of figures 1 and 2 needs to be improved, and the introduction should be rewritten to highlight the novelty of the work and its objective.

I don't see the usefulness of confirming experimental work with a DFT study ;What role did DFT play in the research on the inhibitors a1 and a2?

in the research on the inhibitors a1 and a2?

minor revision needed

Author Response

Reviewer #2

Dear reviewer,

Thank very much for your valuable and interesting comments which shall improve the manuscript.

Comments and Suggestions for Authors

This manuscript must be largely corrected. I pointed out some small or severe problems in the manuscript, as follows.

The resolution of figures 1 and 2 needs to be improved, and the introduction should be rewritten to highlight the novelty of the work and its objective.

Authors’ response: the comment was addressed (cf. Figures 1& 2)

I don't see the usefulness of confirming experimental work with a DFT study ;What role did DFT play in the research on the inhibitors a1 and a2?

 Authors’ response: According to reports, HOMO orbitals provide electrons whereas LUMO orbitals accept electrons. Figures (9,10) displays the optimized molecular structures of the a1 and a2 compounds with the HOMO and LUMO electronic densities that were obtained from theoretical calculations. The HOMO orbitals surrounding the benzyl ring on the a2 molecule, according to the results from Figs. (9,10). For a1, the HOMO orbitals are dispersed throughout the entire molecule. On the other hand, the LUMO orbitals is located on the thiadiazole ring group for a2 and over the entire molecule for a1. This demonstrates the molecules' electron-donating center. The results of DFT demonstrate that a1 and a2 have various electron donners and acceptors sites. This also explains the difference in the inhibition effectiveness for a1 and a2.

Comments on the Quality of English Language

minor revision needed

Authors’ response: the comment was addressed

Reviewer 3 Report

In the present investigation, the authors reported two new corrosion inhibitors for the evaluation of mild steel,i.e., M steel, samples in 0.5M H2SO4 solution.   The two synthesized inhibitors were 2-amino-5-(4-bromobenzyl)-1,3,4-thiadia-zole (a1) and 2-amino-5-(3-nitrophenyl)-1,3,4-thiadiazole (a2). Electrochemical tests such as potentiodynamic polarization, open circuit potential, and electrochemical impedance spectroscopy (EIS) were conducted on the samples with and without the corrosion inhibitors at 25 °C.   The potentiodynamic polarization was conducted to determine the corrosion rate(icorr), the surface coverage of the inhibitor molecules (θ) from the Langmuir adsorption isotherm, and the inhibitor performance (IP%).  Furthermore, the open circuit potential was measured versus time.   Also, EIS was conducted at a frequency range from 100 kHz to 0.1 Hz at an amplitude of 10 mV.   The results of EIS yield values of polarization resistance, solution resistance, charge transfer resistance, and double-layer capacitance from the Nyquist plot and the Bode plot. 

The concentration of the corrosion inhibitors was varied, i.e., 0.0005, 0.001, 0.002, 0.003, 0.004, and 0.005 M, in order to establish a relationship between the inhibitor concentration and the corrosion rate of the mild steel.

In general, one would conclude that Inhibitor a1 performs better than Inhibitor a2 as far as the results of the electrochemical tests.   In addition, the higher the concentration of the inhibitor corresponds to the better the corrosion resistance of the mild steel samples.   This implies that a higher surface coverage of the inhibitor molecules, on the mild steel samples, prevents the charge transfer of the samples.

Eventually, based on the density function Theory (DFT), the structures of a1 and a2 inhibitors can be optimized. 

The following items need to be paid attention to:

1-In the Abstract, on line 5, the symbols " FTIR, C NMR, and H NMR" need to be identified for what they stand for before the abbreviation.

2-Figures 1& 2 are not clear and require redrawing. 

3-Figure 3 needs an improvement in its quality for publication.

Author Response

Reviewer #3

Dear reviewer,

Thank very much for your valuable and interesting comments which shall improve the manuscript.

In the present investigation, the authors reported two new corrosion inhibitors for the evaluation of mild steel,i.e., M steel, samples in 0.5M H2SO4 solution.   The two synthesized inhibitors were 2-amino-5-(4-bromobenzyl)-1,3,4-thiadia-zole (a1) and 2-amino-5-(3-nitrophenyl)-1,3,4-thiadiazole (a2). Electrochemical tests such as potentiodynamic polarization, open circuit potential, and electrochemical impedance spectroscopy (EIS) were conducted on the samples with and without the corrosion inhibitors at 25 °C.   The potentiodynamic polarization was conducted to determine the corrosion rate(icorr), the surface coverage of the inhibitor molecules (θ) from the Langmuir adsorption isotherm, and the inhibitor performance (IP%).  Furthermore, the open circuit potential was measured versus time.   Also, EIS was conducted at a frequency range from 100 kHz to 0.1 Hz at an amplitude of 10 mV.   The results of EIS yield values of polarization resistance, solution resistance, charge transfer resistance, and double-layer capacitance from the Nyquist plot and the Bode plot. 

The concentration of the corrosion inhibitors was varied, i.e., 0.0005, 0.001, 0.002, 0.003, 0.004, and 0.005 M, in order to establish a relationship between the inhibitor concentration and the corrosion rate of the mild steel.

In general, one would conclude that Inhibitor a1 performs better than Inhibitor a2 as far as the results of the electrochemical tests.   In addition, the higher the concentration of the inhibitor corresponds to the better the corrosion resistance of the mild steel samples.   This implies that a higher surface coverage of the inhibitor molecules, on the mild steel samples, prevents the charge transfer of the samples.

Eventually, based on the density function Theory (DFT), the structures of a1 and a2 inhibitors can be optimized. 

The following items need to be paid attention to:

1-In the Abstract, on line 5, the symbols " FTIR, C NMR, and H NMR" need to be identified for what they stand for before the abbreviation.

Authors’ response: the comment was addressed (cf. Abstract)

2-Figures 1& 2 are not clear and require redrawing. 

Authors’ response: the comment was addressed (cf. Figures 1& 2)

3-Figure 3 needs an improvement in its quality for publication.

Authors’ response: the comment was addressed (cf. Figure 3)

Reviewer 4 Report

This work reports the synthesis and the structures of two thiadiazole compounds (a1 and a2) as inhibitors for corrosion control. Authors provide detailed descriptions of synthesis procedures, their effectiveness as mild steel corrosion inhibitors, as well as DFT calculations. The authors claim that a1 and a2 lead to a blockage of charge and mass transfer. Before this paper can be considered for publication, I would like the authors to address the following questions and comments.

1.     Authors need to make sure the English is precise. For example, in abstract “The outcomes show that the inhibition effectiveness rises by raising a1 and a2 concentration of” should be “by raising concentration of a1 and a2”. Another example “a1 and a2 demonstrated the ability to create a protective coating on the M-steel surface against acidic media Surprisingly, negative sign…” there should be a “.” between media and surprisingly.

2.     In the introduction, the authors mention thiadiazole compounds as effective inhibitors, which I believe is the motivation of their study. Therefore, it is necessary to compare the effectiveness of a1 and a2 with other thiadiazole derivatives in the discussion section or the conclusion to highlight the importance of this work.

3.     In figure 1, the numbers in the figures are too small for the readers to recognize.

4.     In the conclusion “Theoretical studies based on DFT agree with research observations demonstrating the efficacy of a1 and a2 as corrosion additives.” Can authors use results in figures 9 and 10 to explain the difference of effectiveness for a1 and a2. I know authors mention that DFT results can explain this briefly, but I would like to see some detailed explanations.

5.     For the SI, did authors mention “figure 2” in the main text? also why is this SI figure labelled as figure 2?

Authors need to make sure the English is precise. For example, in abstract “The outcomes show that the inhibition effectiveness rises by raising a1 and a2 concentration of” should be “by raising concentration of a1 and a2”. Another example “a1 and a2 demonstrated the ability to create a protective coating on the M-steel surface against acidic media Surprisingly, negative sign…” there should be a “.” between media and surprisingly.

Author Response

Reviewer #4

Dear reviewer,

Thank very much for your valuable and interesting comments which shall improve the manuscript.

Comments and Suggestions for Authors

This work reports the synthesis and the structures of two thiadiazole compounds (a1 and a2) as inhibitors for corrosion control. Authors provide detailed descriptions of synthesis procedures, their effectiveness as mild steel corrosion inhibitors, as well as DFT calculations. The authors claim that a1 and a2 lead to a blockage of charge and mass transfer. Before this paper can be considered for publication, I would like the authors to address the following questions and comments.

  1. Authors need to make sure the English is precise. For example, in abstract “The outcomes show that the inhibition effectiveness rises by raising a1 and a2 concentration of” should be “by raising concentration of a1 and a2”. Another example “a1 and a2 demonstrated the ability to create a protective coating on the M-steel surface against acidic media Surprisingly, negative sign…” there should be a “.” between media and surprisingly.

Authors’ response: The comment was addressed

  1. In the introduction, the authors mention thiadiazole compounds as effective inhibitors, which I believe is the motivation of their study. Therefore, it is necessary to compare the effectiveness of a1 and a2 with other thiadiazole derivatives in the discussion section or the conclusion to highlight the importance of this work.

Authors’ response: Thank you for your precious comments.  (Page 10 lines 1-7)  See Table 4

  1. In figure 1, the numbers in the figures are too small for the readers to recognize.

Authors’ response: The comment was addressed (cf. Figure 1)

  1. In the conclusion “Theoretical studies based on DFT agree with research observations demonstrating the efficacy of a1 and a2 as corrosion additives.” Can authors use results in figures 9 and 10 to explain the difference of effectiveness for a1 and a2. I know authors mention that DFT results can explain this briefly, but I would like to see some detailed explanations.

Authors’ response: The comment was addressed (cf. Theoretical studies)

  1. For the SI, did authors mention “figure 2” in the main text? also why is this SI figure labelled as figure 2?

Authors’ response: SI Figures has been transferred to Figure 2

Comments on the Quality of English Language

Authors need to make sure the English is precise. For example, in abstract “The outcomes show that the inhibition effectiveness rises by raising a1 and a2 concentration of” should be “by raising concentration of a1 and a2”. Another example “a1 and a2 demonstrated the ability to create a protective coating on the M-steel surface against acidic media Surprisingly, negative sign…” there should be a “.” between media and surprisingly.

Authors’ response: The comment was addressed

Round 2

Reviewer 4 Report

I am satisfied with the updated manuscript and can now recommend it.